# Electroreductive alkylations of (hetero) arenes with carboxylic acids

Bing Wang[1], Xianshuai Huang[1], Huihua Bi[1] & Jie Liu [1,2] ✉

Carboxylic acids are widely available and generally inexpensive from abundant biomass feedstocks, and they are suitable and generic coupling partners in synthetic chemistry. Reported herein is an electroreductive coupling of stable and versatile carboxylic acids with (hetero)arenes using protons as the hydrogen source. The application of an earth-abundant titanium catalyst has significantly improved the deoxygenative reduction process. Preliminary mechanistic studies provide insights into the deoxygenative reduction of in-situ generated ketone pathway, and the intermediacy generation of ketyl radical and alkylidene titanocene. Without the necessity of pressurized hydrogen or stoichiometric hydride as reductants, this protocol enables highly selective and straightforward synthesis of various functionalized and structurally diverse alkylbenzenes under mild conditions. The utility of this reaction is further demonstrated through practical and valuable isotope incorporation from readily available deuterium source.

Direct alkylation of simple (hetero)arenes belongs to an ideal transformation to construct a carbon–carbon bond involving an aromatic moiety[1]. Conventional Friedel–Crafts alkylation, using an alkylating agent such as alkyl halide through an electrophilic aromatic substitution, has been acknowledged as one of the most fundamental methods (Fig. 1a)[2]. However, it suffers from the limitations of low chemo- and regioselectivities due to the undesired over-alkylation and carbocation rearrangement. Although a combination of Friedel–Crafts acylation[3] and deoxygenative reduction[4–15] can avoid these problems, the multistep manipulations and harsh reductive conditions still restrict their broad and practical applications. Carboxylic acid represents a class of cost-effective, widely available, and structurally diverse feedstock in synthetic organic chemistry[16–22]. The deoxygenative alkylation using carboxylic acids as electrophiles provides a rewarding direction to alleviate the reliance on classic halide chemistry[23]. Pioneered by Gribble, the possibility of direct deoxygenative arylation of trifluoroacetic acid was validated in 1985 (Fig. 1b)[24]. However, the harsh conditions involving borohydrides in great excess and limited scopes stimulated a surge of following research efforts. Later, Sakai described the deoxygenative functionalization of substituted benzoic acids, in which reductively generated silyl ethers by silane served as benzyl electrophiles in the presence of an indium catalyst[25,26]. Beller and his colleagues demonstrated a hydrogenative alkylation of substituted indoles using a wide range of carboxylic acids based on the Co/triphos system[27]. Despite these achievements, the use of sensitive hydride reductants or pressurized hydrogen can potentially result in serious safety issues, elaborate experimental setups as well as high costs, which limits a wide range of applications in synthetic chemistry.

Electrochemistry, in which electrons (e⁻) and protons (H⁺) can be used directly as sustainable and safe redox equivalent and hydrogen source, offers such an opportunity [28–39]. To our knowledge, the application of carboxylic acids as coupling partners in decarboxylative or dehydroxylative functionalizations has been intensively explored in electrochemical synthesis[40–42]. In contrast to these elegant advances, electrochemical deoxygenative functionalization of carboxylic acids through complete deletion of the carboxylic oxygen remains limited, which holds intricate challenges in several aspects. The medium to low electrophilicity of a carboxyl group makes it often thermodynamically and kinetically inert[43–45]. In addition, the selectivity control of the carbonyl reduction remains challenging owing to the formation of alcohol through direct hydrogenation[46–49] as well as methylene product through deoxygenative reduction[50]. Especially towards the desired methylene target constitutes a particular challenge since a higher negative reductive potential is required. Moreover, the

[1]College of Chemistry and Chemical Engineering, State Key Laboratory of Chemo/Biosensing and Chemometrics, Hunan University, 410082 Changsha, China. [2]Greater Bay Area Institute for Innovation, Hunan University, 511300 Guangzhou, China. ✉e-mail: jieliu@hnu.edu.cn

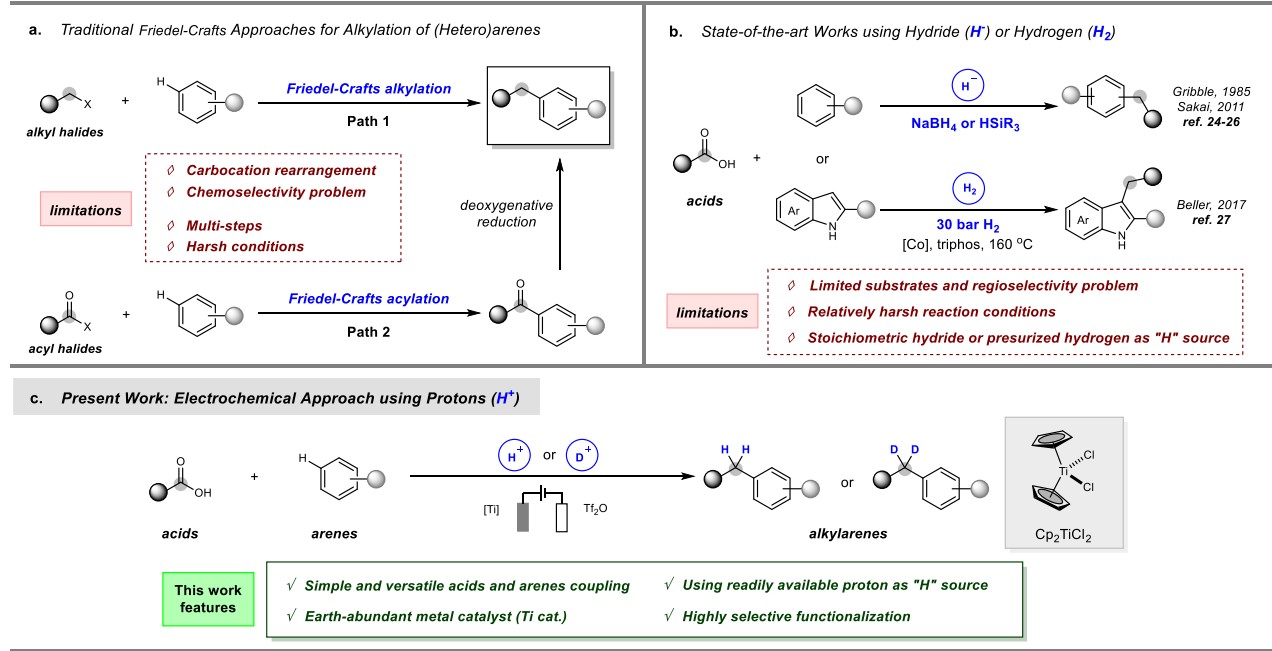

**Fig. 1 | Alkylations of (hetero)arenes using different alkylating reagents and hydrogen sources. a** Traditional Friedel–Craft methods for alkylations of (hetero)arenes. **b** Reductive alkylations of (hetero)arenes with carboxylic acids using hydrides or hydrogen. **c** Electroreductive alkylations of (hetero)arenes with carboxylic acids using protons (this work).

## Table 1 | Effects of reaction parameters[a]

| Entry | Deviation from standard condition | Yield of 3[b] | Yield of 3a[b] |
|---|---|---|---|
| 1 | None | 98 | 0 |
| 2 | Cp*₂TiCl₂ instead of Cp₂TiCl₂ | 54 | Trace |
| 3 | CpTiCl₃ instead of Cp₂TiCl₂ | 66 | 11 |
| 4 | Cp*TiCl₃ instead of Cp₂TiCl₂ | 50 | 25 |
| 5 | Cp₂ZrCl₂ instead of Cp₂TiCl₂ | 54 | 13 |
| 6 | DMAc or DMSO instead of MeCN | 0 | 0 |
| 7 | Ni (–) instead of Pt (–) | 52 | 4 |
| 8 | Graphite plate (–) instead of Pt (–) | 63 | 0 |
| 9 | Fe (+) instead of Zn (+) | 0 | 89 |
| 10 | Al (+) instead of Zn (+) | 36 | 42 |
| 11 | 1 bar H₂ instead of electrolysis | 0 | 16 |
| 12 | No Cp₂TiCl₂ | 67 | 6 |
| 13 | No TFA | 37 | 12 |
| 14 | No Tf₂O | 0 | 0 |
| 15 | No electrolysis | 0 | 85 |

Cp cyclopentadienyl, Cp* pentamethylcyclopentadienyl.
[a] Reaction condition: propionic acid **1** (1.5 mmol), n-butoxybenzene **2** (0.3 mmol), Cp₂TiCl₂ (0.03 mmol), Tf₂O (0.6 mmol), TBABF₄ (0.6 mmol), TFA (2.0 mL) and MeCN (2.0 mL) in an undivided cell with Pt cathode and Zn anode, constant current 30 mA, 70 °C, 3.5 h, N₂.
[b] NMR yield using CH₂Br₂ as an internal standard.

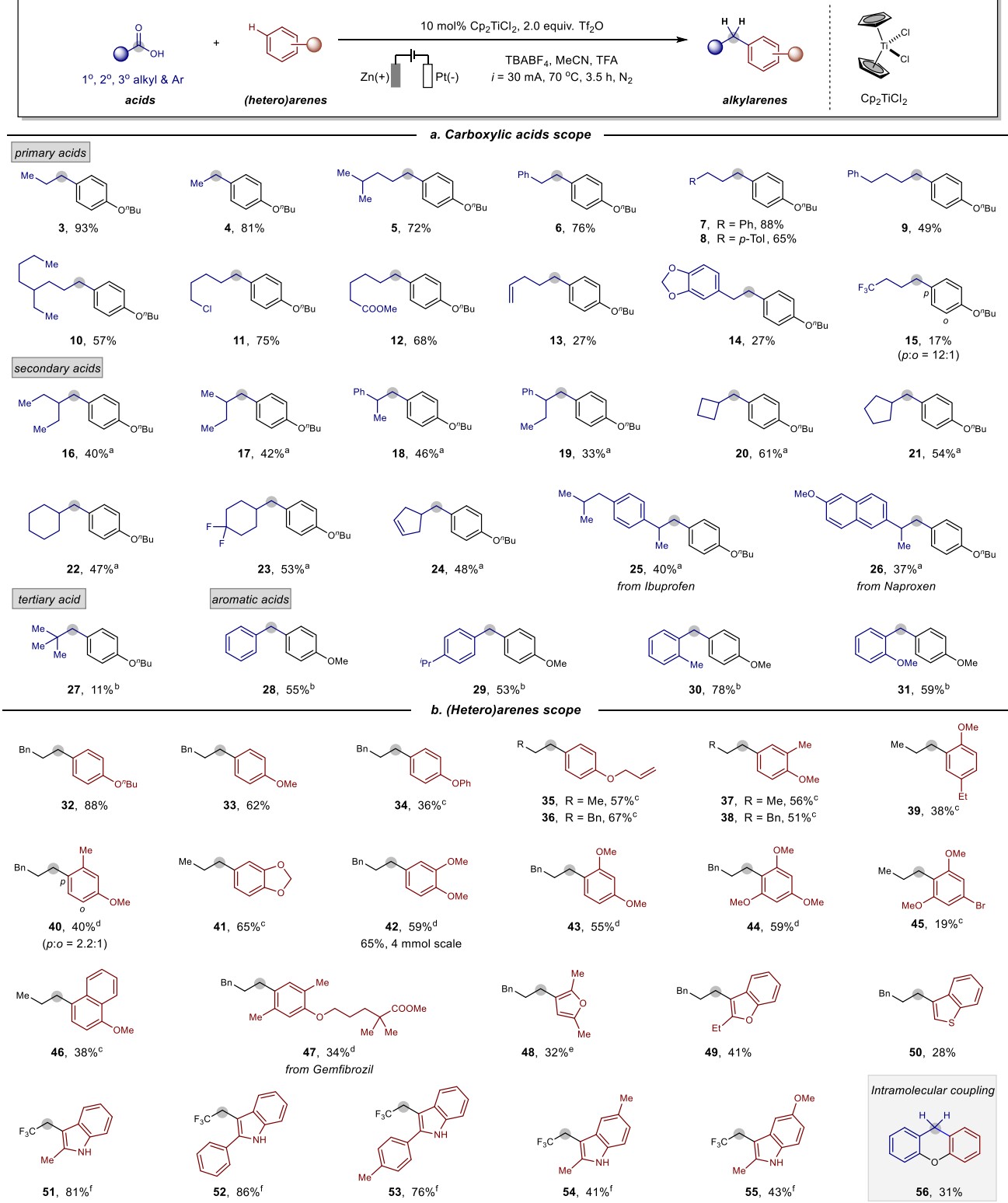

**Fig. 2 | Substrate scope of carboxylic acids and (hetero)arenes.** Reaction condition as shown in Table 1. [a]60 °C, 5 h. [b]Tf₂O (0.675 mmol), TFA (3.0 mL), MeCN (1.0 mL), 50 °C, 6 h. [c]Tf₂O (0.675 mmol), 60 °C, 6 h. [d]Tf₂O (0.3 mmol), r.t., 6 h. [e]TFA (1.0 mL), MeCN (2.0 mL), r.t., 6 h. [f]10 mol% Cp*₂TiCl₂ and diphenyl thiourea instead of Cp₂TiCl₂ and Tf₂O, 90 °C.

carboxylic proton results in a hydrogen evolution reaction as the thermodynamic preference on the cathode competes with the desired carbonyl reduction process[51–53]. To date, a general electrosynthesis of alkylbenzenes via deoxygenative coupling of carboxylic acids has not been established, which is a significant but challenging goal in organic synthesis.

Based on our persistent interest in electrohydrogenations[54], we describe herein a mild and efficient electrochemical deoxygenative

functionalizations of carboxylic acids mediated by an earth-abundant titanium catalyst (Fig. 1c). This method allows for the direct alkylations of (hetero)arenes to produce various functionalized and structurally diverse alkylbenzenes using protons as the hydrogen source. Specifically, a significant advantage of this electrochemical reaction is the highly practical and valuable chemo-divergent isotope incorporation into the alkylbenzenes from a readily available deuterium source.

## Results

### Optimization reaction conditions

In light of these challenges and opportunities, our studies were initiated by screening of reaction parameters for the electrochemical deoxygenative coupling of propionic acid **1** with *n*-butoxybenzene **2** (Table 1). The model reaction was carried out in an undivided cell equipped with a platinum cathode and a zinc anode in trifluoroacetic acid and acetonitrile mixture. Inspired by previous specific activity for deoxygenative couplings[55–59], the earth-abundant titanium complex (Cp$_2$TiCl$_2$) was used as a precatalyst. Under the optimal condition, the desired *n*-propylation product **3** was obtained in an excellent 98% yield (Table 1, entry 1) with high regioselectivity (>50:1) at *para*-position of **2**. The application of other commercially available Ti and Zr complexes such as Cp*$_2$TiCl$_2$, CpTiCl$_3$, Cp*TiCl$_3$, and Cp$_2$ZrCl$_2$ could afford the desired product **3** but in moderate yield (entries 2–5). In addition, the corresponding acylating side-product **3a** was observed in 11–25% yields with these catalysts. The reactions did not proceed when DMAc or DMSO were employed as the solvent (entry 6). The choice of electrode material significantly impacts the efficiency of the electrochemical transformations. Other electrodes, such as Ni, graphite plate as cathodes, or Fe, Al as anodes, led to low or no catalytic activity in the model reaction (entries 7–10). As a comparison, under the pressure of hydrogen gas instead of electrolysis, the reaction did not provide any product (entry 11). In the absence of Cp$_2$TiCl$_2$ as a deoxygenating

reagent, the transformation proceeded with decreased efficiency (entry 12). Moreover, using trifluoroacetic acid (TFA) as a co-solvent provided sufficient proton source and acidic condition for this transformation (entry 13). Notably, triflic anhydride (Tf$_2$O) is essential to activate carboxylic acids and generate highly potent triflate electrophiles in this reaction (entry 14). The control experiment demonstrated that electricity is necessary for the deoxygenation process (entry 15).

### Substrate scope

After establishing the optimized reaction conditions, the generality of this electrochemical alkylation was studied (Fig. 2). First, we tested various substituted and functionalized aliphatic and aromatic carboxylic acids to produce the corresponding alkylbenzenes. In addition to propionic acid **1**, various short and longer-chain aliphatic carboxylic acids performed well, forming the corresponding products **4–10** in moderate to good yields (49–88%). Specifically, carboxylic acids with the chlorine, ester, olefin, and 1,3-benzodioxole functionalities underwent this transformation smoothly, and their successful conversion to substituted arenes **11–14** expands the scope of this reaction. The application of trifluorobutyric acid as a substrate led to the corresponding alkylating product **15**, albeit with decreased yield and regioselectivity (12:1). Besides primary carboxylic acids, secondary carboxylic acids also proved compatible under this electrochemical condition. A wide variety of structurally diverse long-chain and cyclic acids performed well, and synthetic, useful yields of **16–24** were obtained. Gratifyingly, pharmaceutical molecules such as Ibuprofen and Naproxen were found to be suitable substrates to afford the corresponding alkylbenzenes **25** and **26** in moderate yields. Moreover, the use of pivalic acid as a substrate led to an 11% yield of the desired product **27**, which is likely due to the challenging deoxygenative process caused by the high steric hindrance of the substrate. To our delight, beyond aliphatic carboxylic acids as substrates, the present system could be effectively applied to aromatic acids affording the corresponding diarylmethanes **28–31** in 53–78% yields. Notably, unless otherwise mentioned, the deoxygenative alkylations were highly selectively performed at the *para*-position of *n*-butoxybenzene or anisole with more than 50:1 regioselectivity.

Next, we examined the substrate scope by employing structurally diverse (hetero)arenes. Aryl ethers with different substitutions such as methyl, phenyl, and allyl groups were well compatible with this methodology and gave the desired products **32–36** in 36–88% yields and excellent regioselectivities (>50:1). Moreover, a range of disubstituted arenes proved to be good coupling partners, providing the corresponding alkylarenes **37–43** in moderate to good yields (38–65%). Notably, moderate regioselectivity of 2.2:1 was observed for product **40** when 3-methylanisole was applied as a substrate. Additionally, a scale-up reaction with 1,2-dimethoxybenzene as a substrate gave a 65% yield of the corresponding product **42**. Benzenes bearing trisubstitutions such as 1,3,5-trimethoxybenzene **44** and 1-bromo-3,5-dimethoxybenzene **45** were also found to be suitable substrates and underwent these alkylations smoothly. Similarly, 1-methoxynaphthalene provided a moderate yield of the desired product **46**. The Gemfibrozil derivative, a pharmaceutical used to reduce cholesterol and triglycerides, also furnished the desired products **47** in 34% yields. Interestingly, heteroaromatics such as substituted furan **48**, benzofuran **49**, and benzothiophene **50** participated in this transformation, highlighting the broad substrate scope of this protocol. Moreover, the deoxygenative coupling of various indoles with trifluoroacetic acid also performed efficiently and gave the corresponding trifluoroethylating products **51–55** in 41–86% yields without the necessity of triflic anhydride. In addition to intermolecular couplings, the use of 2-phenoxybenzoic acid as a substrate led to xanthene **56** in 31% yield through an intramolecular reductive cyclization.

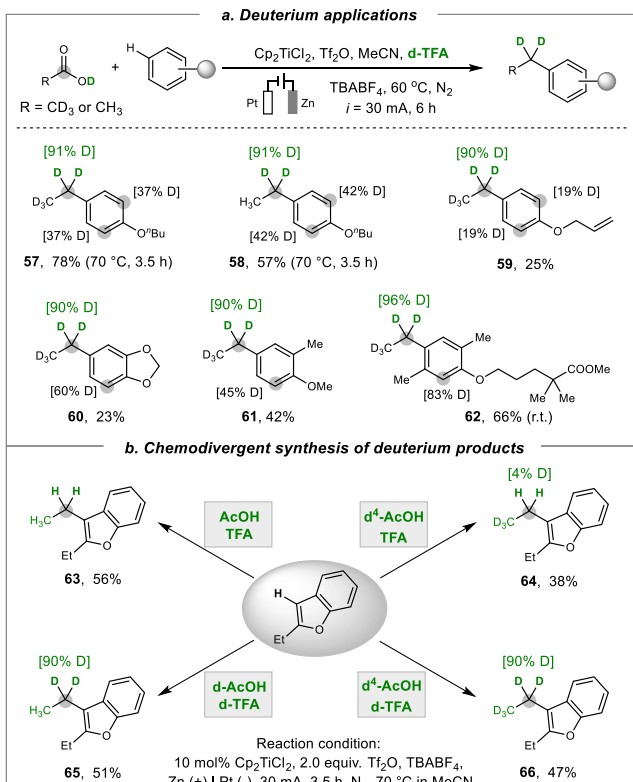

**Fig. 3 | Synthetic electrochemical deuterium labeling transformations.**
**a** Deuterium applications. **b** Chemodivergent synthesis of deuterium products.

## Deuterium applications

Organic molecules labeled with hydrogen isotopes are highly desirable in analytical science and pharmaceutical chemistry[60]. The replacement of hydrogen by deuterium (D) at the benzylic position often gives rise to improved metabolic stability and bioactivity in drug discovery. Here the present electrocatalytic deoxygenative reduction provides a practical and cost-effective approach to benzylic deuterium labeling (Fig. 3a). The application of deuterated acetic acids with different arenes furnished valuable products 57−62 in 23−78% yields with 90−96% deuterium incorporations. Notably, in some cases, the aromatic C−H position was also partly deuterated at the *ortho*-position of ether functionality, as supported by natural population analysis (Supplementary Fig. 12). Remarkably, this catalytic system also allows for a chemo-divergent synthesis of different deuterium products starting from 2-ethylbenzofuran (Fig. 3b). The selectivity control was achieved

by using different acetic acids (AcOH, d-AcOH and d[4]-AcOH) and trifluoroacetic acids (TFA and d-TFA) as deuterium sources. The use of non-deuterated AcOH and TFA under optimal conditions led to a 56% yield of ethylative product 63, whereas the $CD_3$-labled product 64 is obtained in 38% yield with d[4]-AcOH. In addition, the application of d-TFA with two different d-AcOH and d[4]-AcOH resulted in a d[2]-methylene ($-CD_2-$) product 65 and a fully deuterated ethyl ($CD_3CD_2-$) product 66 with excellent deuterium incorporation. These results also indicate that the hydrogen source for deoxygenative reduction mainly comes from trifluoroacetic acid.

## Mechanistic investigations

To gain insights into this electroreductive coupling, a series of mechanistic experiments were performed (Fig. 4). Firstly, the interactions between the carboxylic acid, arene, and triflic anhydride were

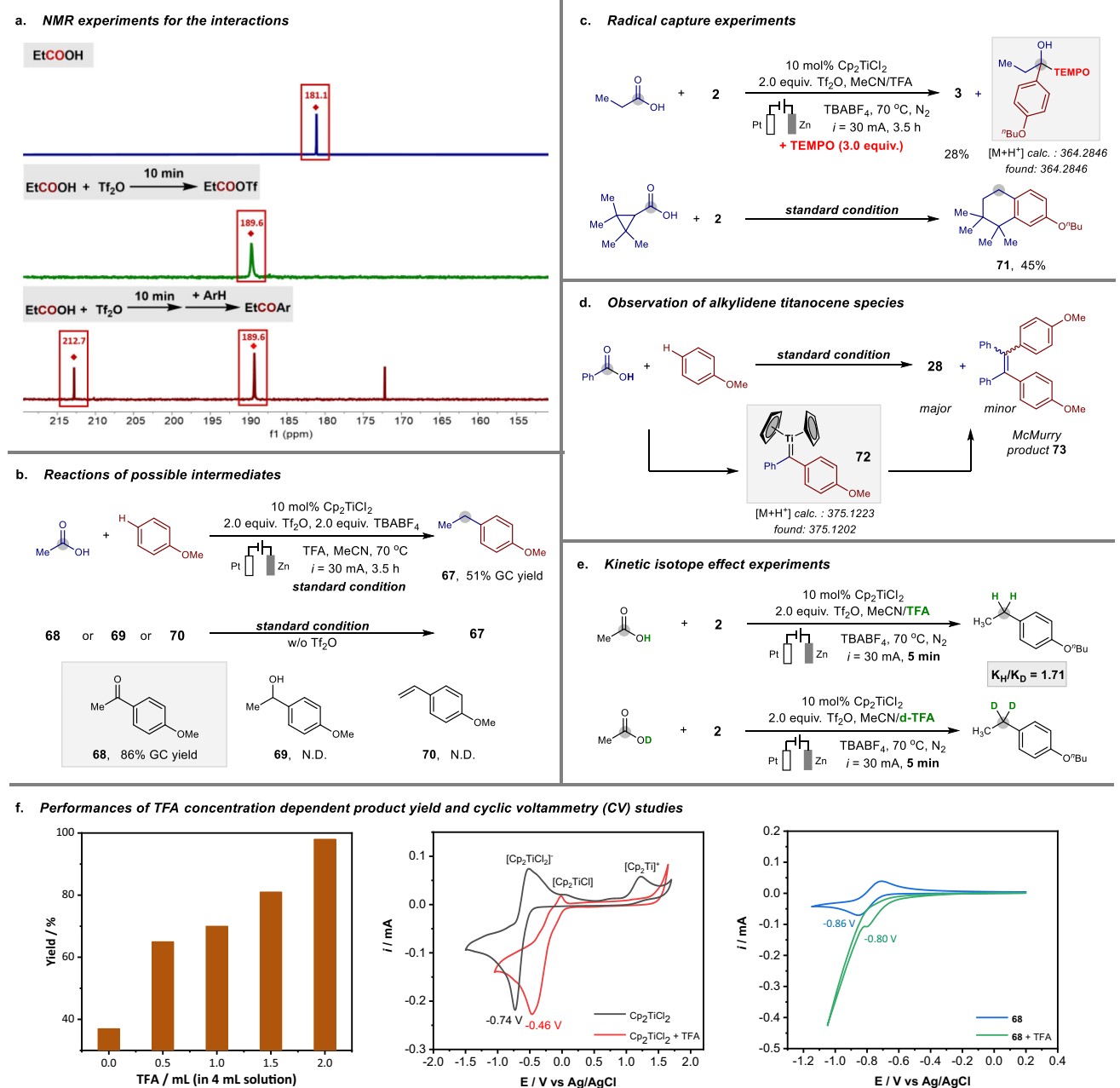

**Fig. 4 | Mechanistic studies. a** NMR experiment for the interactions. **b** Reactions of possible intermediates. **c** Radical capture experiments. **d** Observation of alkylidiene titanocene intermediates. **e** Kinetic isotope effect experiments. **f** Performances of TFA concentration-dependent product yield and cyclic voltammetry (CV) studies.

monitored by $^{13}$C NMR. As shown in Fig. 4a, a significant chemical shift from 181.1 to 189.6 ppm was observed after the addition of Tf$_2$O to the propionic acid **1** solution, indicating the rapid formation of a highly electrophilic triflate intermediate. Subsequently, the addition of *n*-butoxybenzene **2** resulted in the observation of a chemical shift downfield to 212.7 ppm, suggesting a ketone intermediate generation. Next, the catalytic reactions of possible intermediates were performed under standard conditions (Fig. 4b). The model hydrogenative coupling of acetic acid with anisole gave the desired 4-ethylanisole **67** in 51% yield. Under the standard conditions, deoxygenative reduction of 4-methoxyacetophenone **68** led to 86% yield, confirming that the in-situ generated ketone by Friedel–Crafts acylation is the key intermediate. This ketone intermediate then participates in the subsequent electrochemical hydrogenolysis[12–14]. We also investigated the potential intermediacy of an alcohol which could be produced by ketone hydrogenation and corresponding alkene generated by dehydration of the alcohol[61]. However, the application of 1-(4-methoxyphenyl)ethanol **69** and 4-methoxystyrene **70** did not afford the desired ethylating product, indicating the proposed alcohol and alkene are not intermediates in this transformation.

Next, more experiments were conducted to provide an understanding of the reaction mechanism. We firstly performed radical capture experiments to explore the reaction patterns of carboxylic acids (Fig. 4c). The addition of 2,2,6,6-tetramethyl-1-piperidinyloxy (TEMPO) led to a significant decreasing yield of the desired product **3** and the adduct of TEMPO with a ketyl radical was observed by high-resolution mass. Interestingly, the use of 2,2,3,3-tetramethylcyclopropanecarboxylic acid as a substrate generated a six-membered bicycle **71** in 45% yield through a sequence of cyclopropane ring opening and intramolecular cyclization. These results indicate a ketyl radical species formed via single-electron reduction of a ketone could be related[62]. In addition, high-resolution mass analysis of the hydrogenative coupling of benzoic acid with anisole suggested the involvement of a titanium carbene intermediate **72** in this transformation[63], and a tetrasubstituted alkene **73** by McMurry reaction was also observed as a minor product (Fig. 4d)[64–66]. Moreover, the kinetic isotope effect experiments were performed using deuterated and non-deuterated reagents (Fig. 4e). The $K_H/K_D$ value was found to be 1.71, revealing a primary kinetic isotope effect for the hydrogen transfer process.

In this reaction, trifluoroacetic acid (TFA) is not only used as the proton source for deoxygenative electroreduction, but also provides suitable acidic conditions to achieve the desired transformation (Fig. 4f). We found that the concentration of TFA significantly impacts the reaction efficiency. In the absence of TFA, the model reaction proceeded with low activity, whereas increasing TFA concentration led to obvious higher yields of desired product **3**. Additionally, cyclic voltammetry (CV) experiments were investigated to acquire a further understanding of the interaction of TFA with Cp$_2$TiCl$_2$ and ketone intermediate. The CV profile of Cp$_2$TiCl$_2$ at 100 mV/s showed reversible Ti redox couples (black line)[67,68]. The addition of TFA to the titanium complex solution led to an obvious positive shift in the reduction potential (red line). On the other hand, adding TFA to a ketone intermediate **68** solution also resulted in the observation of heightened current response and a more positive peak shift of the reduction potential (blue line and green line). Moreover, the anodic wave corresponding to oxidation of **68** is not observed on the return scan. Such observations are indicative of a more feasible reduction of titanium catalyst and an irreversible ketone electroreduction to ketyl radical in the presence of TFA.

Based on these results, a possible mechanism for this electroreductive coupling is proposed in Fig. 5. Initial electrophilic activation of a carboxylic acid with triflic anhydride leads to the formation of a highly electrophilic triflate, which is prone to attack by a (hetero)arene to furnish a ketone **A**. This intermediate is reduced through a single electron transfer process to afford a ketyl radical **B** which has been confirmed by radical trapping experiments (Fig. 4c). Subsequently, reductive deoxygenation in **B** takes place and generates a carbene **C**[69]. There are two pathways for the subsequent transformation. In the presence of a titanium catalyst (path 1), the carbene **C** is trapped by the Cp$_2$Ti(II) species, which is generated from the reduction of Cp$_2$TiX$_2$ (X = Cl or OTFA) on the cathode, to provide an alkylidene titanocene **D**. This key intermediate was observed by the high-resolution mass (Fig. 4d). Protonation of **D** yields the desired alkylating product and regenerates Cp$_2$TiX$_2$. Alternatively, this electroreductive coupling also provided the desired product in moderate yield (67%) in the absence of a titanium catalyst. Therefore, a direct electrochemical hydrogenolysis (path 2) through stepwise protons and electrons transfer on the cathode might also deliver the corresponding hydrocarbons[70].

## Discussion

In summary, we have developed a selective and efficient electrochemical deoxygenative alkylation of (hetero)arenes using stable and versatile carboxylic acids. The reaction is significantly improved by an earth-abundant titanium catalyst as the deoxygenative reagent. Mechanistic investigations revealed that the in-situ generated ketone, identified as the key intermediate, is involved in the subsequent electrochemical deoxygenative reduction. The present method exhibits a broad substrate scope with good functional group compatibility and is amenable to valuable isotope incorporation from readily available protons. Given the broad availability and diversity of carboxylic acids and (hetero)arenes, we anticipate this protocol will find potential utility and facilitate further explorations in synthetic chemistry.

## Methods

### Representative procedure for the synthesis of compound 3

A dried 10 mL glass tube equipped with a magnetic stirring bar was added Cp$_2$TiCl$_2$ (0.03 mmol), *n*-butoxybenzene **2** (0.3 mmol), propionic acid **1** (1.5 mmol), TBABF$_4$ (0.6 mmol), Tf$_2$O (0.6 mmol), CF$_3$COOH (2.0 mL) and MeCN (2.0 mL). The reactor was equipped with a Zn electrode (2 cm × 1.5 cm × 0.05 cm) as the anode and a Pt electrode (2 cm × 1 cm × 0.02 cm) as the cathode. The reaction was bubbled with N$_2$ for 5 min. Then the mixture was electrolyzed under a constant current of 30 mA for 3.5 h at 70 °C. After the reaction was completed, the reaction solvent was diluted with 40 mL ethyl acetate and washed with sat. NaHCO$_3$ (aq.) solution three times, dried over Na$_2$SO$_4$ and organic layers were combined and concentrated in vacuo. The resulting residue was purified by silica gel flash chromatography to give the product **3** in a 93% isolated yield.

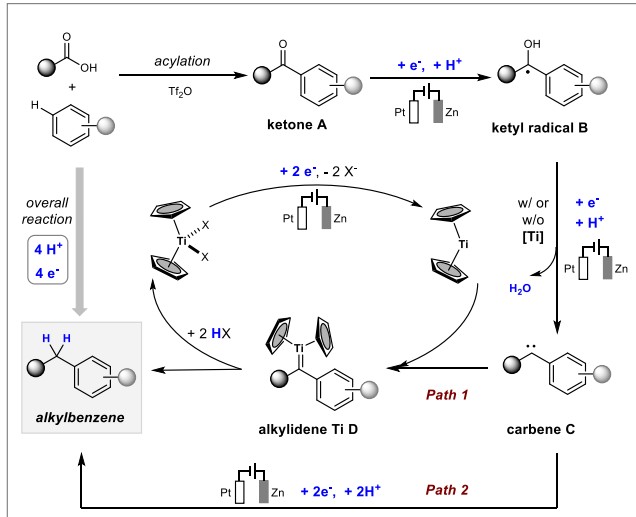

**Fig. 5 | Proposed mechanism.**

## Data availability

The data reported in this paper are available within the article and its Supplementary Information files. All data are also available from the corresponding author upon request.

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

## Acknowledgements
The authors are grateful for the natural population analysis by Prof. Shuanglin Qu's and financial support from the Natural Science Foundation of China (22301073), the Natural Science Foundation of Hunan Province (2021JJ40043, 2021RC3056), and the Fundamental Research Funds for the Central Universities.

## Author contributions
B.W. and J.L. conceived and designed the project. B.W., X.H., and H.B. conducted the experiments. B.W. and J.L. wrote the manuscript. All authors contributed to analyzing the data and editing the manuscript.

## Competing interests
The authors declare no competing interests.
