## [Peer Review File · Nature Communications]

Electroreductive Alkylations of (Hetero)arenes with Carboxylic AcidsREVIEWER COMMENTS

Reviewer #1 (Remarks to the Author):

Liu and coworkers developed an electrochemical approach for alkylations of (hetero)arenes using stable and versatile carboxylic acids. The article describes a sustainable and selective method for synthesizing structurally diverse alkylbenzenes using only electrons and protons as the redox equivalent and hydrogen source. A potentially valuable entry can be easy access to deuterated products from readily available deuterium sources. The authors provide evidence for this mechanism through various experiments, including electrochemical studies and isotopic labeling experiments. As the article is well put together, I recommend publication of this work with some minor revisions.

1. The authors mentioned that titanium catalysts serve a deoxygenation role in the reaction process. How did the author demonstrate this process?
2. In Fig 2, the generation of product 56 through intramolecular cyclization resulted in a relatively low yield (31%). An explanation for this process is required.
3. Did the authors observe the formation of multi-substituted alkyl products? Please comment.
4. In addition to simple carboxylic acids, would it be interesting to test amino acids as substrates, including protected and unprotected amino acids?
5. Since a large amount of trifluoroacetic acid was used as cosolvent, it might compete with the carboxylic acid substrate in the reaction. Did the author observe any trifluoroethylation as a side product?
6. In Fig.3, Besides the deuteration of the carbonyl group, did the author observe deuterium incorporation at the α -position? Did the authors also investigate the possibility of deuterium incorporation at other positions?

Reviewer #2 (Remarks to the Author):

In this paper, Jie Liu et al. reported electrochemical C-H alkylations of (hetero)arenes with carboxylic acids, delivering a new method to construct carbon-carbon bonds. This electroreduction method exhibits a broad substrate scope with good functional group compatibility and the use of earth-abundant titanium catalyst is the key. Moreover, preliminary mechanistic studies are investigated and a possible mechanism is proposed. From the aspect of synthetic chemistry, this paper provides a new method to achieve the direct alkylation of simple (hetero)arenes. In my opinion, this paper appeals to a broad readership, but a "major revision" before acceptance is essential.

1. In Table 1, compound 3' was generated in the absence of electricity with 85% yield (entry 15), so in my opinion, the essence of this electrochemical reaction is the deoxygenative reduction of in-situ generated aryl alkyl ketones, rather than the electrochemical C-H alkylations of (hetero)arenes with carboxylic acids as described by the authors. Furthermore, the yield of compound 3 can reach 67% without Cp_2TiCl_2 , so the authors should not ignore the research process of this kind of electrochemical reaction (such as ChemCatChem 2023, 15, e202300258). Moreover, no product was observed when CH_3CN was replaced by DMAc or DMSO (entry 6), and compound 3 was obtained at 37% yield in the absence of TFA, so whether CH_3CN was involved in this electrochemical reaction?
2. As described by the authors, TFA is the main but not the only hydrogen source, so is the molecular formula of compound 64 in Fig. 3b correctly written? Whether $\text{d}_4\text{-AcOH}$ is also one of the hydrogen sources, the ^1H NMR integral (1.92) of 64 also shows that this

formulation may be incorrect. Deuterium spectroscopy may resolve this question.

3. The results of the radical trapping experiments in Fig. 4c are confusing. What role do they play in the proposed mechanism and how do they generate intermediate Int-B?

4. Anodic sacrifice should exist in this electroreduction reaction and be expressed in the proposed mechanism.

5. Please carefully check the tables, figures and main text, and unify the format of the reaction equations, as there are so many errors, some of which are indicated as follows:

(1) Line 70, product 3'; Table 1, 3', 3'

(2) Table 1, TBABF₄, nBu₄NBF₄

(3) Table 1, i = 30 mA,; Fig. 2, Fig. 3, cc = 30 mA; Fig. 4, i = 30 mA

(4) Fig. 2, there should be spaces before and after the symbol "x"

(5) Line 35, "Despite these achievements"

(6) Line 51, "reductions process"

(7) Line 82, "the generality of this electrochemical alkylations"

(8) Line 90, " this electrochemical conditions"

(9) Line 169, "intemediate 68"

(10) Line 171, " the anodic waves of 68 is..."

(11) Line 188, "we have developed a... alkylations..."

Reviewer #3 (Remarks to the Author):

This manuscript written by Jie Liu and co-workers deals with an electrochemical C-H alkylations of arenes with carboxylic acids as alkylating reagent. Although, at first glance, this work looks good, actually the transformation is not, at least, as novel as the authors claimed. I do not think this manuscript is suitable for publishing on Nat. Commun.

Mechanistically, this work combines two known parts, namely 1) the Tf₂O participated Friedel-Crafts acylation with electron-rich arenes and carboxylic acids to generate alkyl aryl ketones; 2) which are involved in the following Cp₂TiCl₂-mediated electrochemical deoxygenative reduction.

1) In addition, since the deoxygenative reduction is not difficult to occur, the conditions reported in this work are not specific. For example, In Table 1, entry 12, the control experiment carried out still resulted the formation of the desired product 3 in 67% yield in the absence of the catalyst Cp₂TiCl₂, albeit lower than that obtained under their standard conditions. These results questioned the role of Cp₂TiCl₂ played in the catalytic cycle, and the words in the abstract 'The key to success for this transformation is the use of a molecular titanium catalyst for the efficient deoxygenative alkylation of (hetero)arenes' are not scientifically correct.

2) In Fig. 3, the authors commented on line 123, 'Notably, in some cases the aromatic C-H position was also partly deuterated with this method.', the reason for obtaining these 'partly deuterated' results should be provided.

3) Personally, this reviewer DO NOT like the writing style of abstract in this manuscript, which only comments the potential or specious merits but escapes the key mechanistic insight of the reaction, unless the authors may also believe that this work is not that new and nothing relating to the mechanism is worth to be commented in their abstract.

Response letter

Reviewer #1 (Remarks to the Author):

Liu and coworkers developed an electrochemical approach for alkylations of (hetero)arenes using stable and versatile carboxylic acids. The article describes a sustainable and selective method for synthesizing structurally diverse alkylbenzenes using only electrons and protons as the redox equivalent and hydrogen source. A potentially valuable entry can be easy access to deuterated products from readily available deuterium sources. The authors provide evidence for this mechanism through various experiments, including electrochemical studies and isotopic labeling experiments. As the article is well put together, I recommend publication of this work with some minor revisions.

1. The authors mentioned that titanium catalysts serve a deoxygenation role in the reaction process. How did the author demonstrate this process?

Our response: we have two aspects to demonstrate this process:

- (1) Regarding the experimental evidence in this transformation, high-resolution mass analysis of the coupling of benzoic acid with anisole suggested a titanium carbene intermediate involved in this reaction (please also see Fig. 4d in the manuscript). Moreover, a tetrasubstituted alkene by McMurry reaction was also observed as the minor product. These results indicate that titanium catalyst serves as a deoxygenation role in the reaction process.
- (2) In addition, recent elegant reports also support titanium can act as an efficient deoxygenation reagent for the reduction of $-CO-$ to $-CH_2-$ (for recent reviews: *Chem. Soc. Rev.* **2020**, *49*, 6947; *ChemCatChem*, **2022**, *14*, e202200530; *Chem*, **2022**, *8*, 1805 and so on). These seminal works were cited as Ref. 55-59 in the manuscript.

2. In Fig 2, the generation of product 56 through intramolecular cyclization resulted in a relatively low yield (31%). An explanation for this process is required.

Our response: Regarding the intramolecular reaction, an intramolecular acylation was observed to give the cyclic ketone as the side product.

3. Did the authors observe the formation of multi-substituted alkyl products? Please comment.

Our response: we observed trace amount of disubstituted alkylation products (<5%) by GC-MS

and this side product can be avoided under lower reaction temperature.

4. In addition to simple carboxylic acids, would it be interesting to test amino acids as substrates, including protected and unprotected amino acids?

Our response: we tested some free and protected amino acids (please see below), however, these substrates did not work in this electrocatalytic conditions. We have added these unsuccessful substrates in the Supporting Information on page S9.

5. Since a large amount of trifluoroacetic acid was used as cosolvent, it might compete with the carboxylic acid substrate in the reaction. Did the author observe any trifluoroethylation as a side product?

Our response: we did not observe the corresponding trifluoroethylating product in the reaction. To understand this process, the ^{13}C NMR experiments were performed to investigate the interactions. As shown below, we did not observe any new signal for the addition of Tf_2O to the TFA solution. Thus, there might be no interactions of TFA with Tf_2O under this reaction condition. This result was included in the Supporting Information on page S10.

6. In Fig.3, Besides the deuteration of the carbonyl group, did the author observe deuterium incorporation at the α -position? Did the authors also investigate the possibility of deuterium incorporation at other positions?

Our response: besides partly deuterium incorporation at aromatic C-H position (Fig. 3 in

manuscript), we did not observe any deuterium incorporation at the α -position and other positions.

Reviewer #2 (Remarks to the Author):

In this paper, Jie Liu et al. reported electrochemical C-H alkylations of (hetero)arenes with carboxylic acids, delivering a new method to construct carbon-carbon bonds. This electroreduction method exhibits a broad substrate scope with good functional group compatibility and the use of earth-abundant titanium catalyst is the key. Moreover, preliminary mechanistic studies are investigated and a possible mechanism is proposed. From the aspect of synthetic chemistry, this paper provides a new method to achieve the direct alkylation of simple (hetero)arenes. In my opinion, this paper appeals to a broad readership, but a “major revision” before acceptance is essential.

1. (1) In Table 1, compound 3' was generated in the absence of electricity with 85% yield (entry 15), so in my opinion, the essence of this electrochemical reaction is the deoxygenative reduction of in-situ generated aryl alkyl ketones, rather than the electrochemical C-H alkylations of (hetero)arenes with carboxylic acids as described by the authors.

Our response: We made some revisions based on the suggestion from this reviewer:

(1) **The title** has been revised to “**Electroreductive Alkylations of (Hetero)arenes with Carboxylic Acids**”, emphasizing the reduction process and omitting the word “C-H”.

(2) Additionally, We have revised **the abstract** to include mention of the ketone intermediate and the deoxygenative reduction process in this transformation.

(2) Furthermore, the yield of compound 3 can reach 67% without Cp₂TiCl₂, so the authors should not ignore the research process of this kind of electrochemical reaction (such as ChemCatChem 2023, 15, e202300258).

Our response: We mentioned these related works in the manuscript **on page 7 lines 151-152 and cited as the ref. 12-14** in the manuscript.

(3) Moreover, no product was observed when CH₃CN was replaced by DMAc or DMSO (entry 6), and compound 3 was obtained at 37% yield in the absence of TFA, so whether CH₃CN was involved in this electrochemical reaction?

Our response: we think CH₃CN has two functions in this work:

(1) We applied CD₃CN instead of CH₃CN as the solvent to confirm the source of hydrogen. However, no deuterium incorporation was observed in product. Consequently, we postulate that CH₃CN primarily functions as an effective solvent to facilitate the electrochemical deoxygenative process due to its good electric conductivity. In addition, we found DMSO

or DMAc were incompatible with Ti_2O under standard condition.

- (2) Furthermore, according to a related paper (*ChemSusChem* **2019**, *12*, 3166, also cited as ref. 68 in the manuscript), aprotic polar solvent such as CH_3CN was found to be essential for preventing electrode deactivation of Ti catalyst and led to equilibrium shifted towards the active low valence Ti species.

2.As described by the authors, TFA is the main but not the only hydrogen source, so is the molecular formula of compound 64 in Fig. 3b correctly written? Whether $\text{d}_4\text{-AcOH}$ is also one of the hydrogen sources, the ^1H NMR integral (1.92) of 64 also shows that this formulation may be incorrect. Deuterium spectroscopy may resolve this question.

Our response: according to the suggestion, a deuterium spectroscopy experiment was performed. As shown below, we observed approximately 4% D enrichment at the benzylic position, which originated from $\text{d}_4\text{-AcOD}$ as the deuterium donor. **We have corrected this error in the manuscript Fig. 3 and have included the D NMR spectrum in the supporting information on page S109.**

20240424-WB-CHCC13.1.tif

3. The results of the radical trapping experiments in Fig. 4c are confusing. What role do they play in the proposed mechanism and how do they generate intermediate Int-B?

Our response:

- (1) Based on the radical trapping experiments, the ketyl radical resulting from ketone SET reduction was identified by addition of TEMPO. We observed this intermediate (adduct of TEMPO with a ketyl radical) by high-resolution mass spectrometry, as depicted in Fig. 4c of the manuscript.
- (2) The Int-B was proposed to arise from the interaction between titanium complex and the ketyl radical species. Although we made a lot of attempts (such as HRMS and CV) to substantiate this interaction, **we do not have solid experimental evidence for the formation of the titanium-coordinated ketyl radical Int-B.**

- (3) Taking into account the reviewer's concern and our experimental findings, we have revised the mechanism presented in Fig. 5, **where titanium-coordinated ketyl radical Int-B is no longer mentioned in the updated mechanism.** Additionally, we have included an alternative pathway in the absence of Ti catalyst (referred as path 2).

4. Anodic sacrifice should exist in this electroreduction reaction and be expressed in the proposed mechanism.

Our response: the anodic sacrifice (Zn) is presented in the mechanism part (Fig. 5).

5. Please carefully check the tables, figures and main text, and unify the format of the reaction equations, as there are so many errors, some of which are indicated as follows:

(1) Line 70, product 3'; Table 1, 3', 3'

Our response: we have corrected 3', 3' to 3'.

(2) Table 1, TBABF₄, nBu₄NBF₄

Our response: we have corrected nBu₄NBF₄ to TBABF₄.

(3) Table 1, *i* = 30 mA; Fig. 2, Fig. 3, *cc* = 30 mA; Fig. 4, *i* = 30 mA

Our response: we have corrected all *cc* to *i*.

(4) Fig. 2, there should be spaces before and after the symbol "x"

Our response: we have corrected nBu₄NBF₄ to TBABF₄.

(5) Line 35, "Despite these achievements"

Our response: we have corrected "achievements" to "achievements".

(6) Line 51, "reductions process"

Our response: we have corrected "reductions" to "reduction".

(7) Line 82, "the generality of this electrochemical alkylations"

Our response: we have corrected "alkylations" to "alkylation".

(8) Line 90, " this electrochemical conditions"

Our response: we have corrected "conditions" to "condition".

(9) Line 169, "intemediate 68"

Our response: we have corrected "intemediate" to intermediate.

(10) Line 171, " the anodic waves of 68 is..."

Our response: we have corrected "waves" to "wave".

(11) Line 188, "we have developed a... alkylations..."

Our response: we have corrected "alkylations" to "alkylation".

Reviewer #3 (Remarks to the Author):

This manuscript written by Jie Liu and co-workers deals with an electrochemical C-H alkylations of arenes with carboxylic acids as alkylating reagent. Although, at first glance, this work looks good, actually the transformation is not, at least, as novel as the authors claimed. I do not think this manuscript is suitable for publishing on Nat. Commun. Mechanistically, this work combines two known parts, namely 1) the Tf₂O participated Friedel-Crafts acylation with electron-rich arenes and carboxylic acids to generate alkyl aryl ketones; 2) which are involved in the following Cp₂TiCl₂-mediated electrochemical deoxygenative reduction.

Our response: Thanks for the comments. We highlight present work in several aspects as following:

- (1) Deoxygenative functionalization of carboxylic acid is highly attractive in synthetic chemistry. Mechanistically, there are two pathways for this transformation (*Angew. Chem. Int. Ed.* 2022, 61, e202112770). The carboxylic acid can be reduced to the aldehyde, which subsequently undergoes deoxygenative functionalization to form the desired product. Alternatively, direct acylative functionalization may also take place initially, followed by deoxygenative reduction to afford the desired product. Given the relatively low electrophilicity of carboxyl group, the addition of extra additives (e.g. Brønsted and Lewis acid, PPh₃, NHPI, Tf₂O, Boc₂O, etc.) are classical strategies to address thermodynamically and kinetically inert problem. In present work, **mechanistic studies revealed that the second pathway (the acylation-reduction route) is associated with Tf₂O as the activating reagent.**
- (2) The synthesis of complex target molecules from versatile and abundant feedstocks is always highly desirable and economical. Our method allows for the production of various functionalized and structurally diverse alkylbenzenes using readily available carboxylic acids and (hetero)arenes. A broad scope of carboxylic acids, including primary, secondary, tertiary and aromatic acids are suitable substrates as the alkylating reagents under these reaction conditions. Moreover, **this method also provides an alternative and improved alkylation approach to address conventional Friedel–Crafts limitations, such as over-alkylation, carbocation rearrangement and multistep manipulations.**
- (3) In this work, we describe an electrochemical approach for deoxygenative alkylations of

(hetero)arenes, in which electrons (e^-) and protons (H^+) are used directly as sustainable and safe redox equivalents and a hydrogen source. This process avoids the use of hazardous pressurized H_2 or sensitive hydride reductants, thus providing a mild, safe, and easily manipulated method for deoxygenative reduction. Additionally, **the utility of this reaction is further demonstrated through practical and valuable isotope incorporation from readily available deuterium source.**

1. In addition, since the deoxygenative reduction is not difficult to occur, the conditions reported in this work are not specific. For example, In Table 1, entry 12, the control experiment carried out still resulted the formation of the desired product 3 in 67% yield in the absence of the catalyst Cp_2TiCl_2 , albeit lower than that obtained under their standard conditions. These results questioned the role of Cp_2TiCl_2 played in the catalytic cycle, and the words in the abstract ‘The key to success for this transformation is the use of a molecular titanium catalyst for the efficient deoxygenative alkylation of (hetero)arenes’ are not scientifically correct.

Our response:

(1) In general, there are two pathways for the reduction of an unsaturated $C=O$ bond. Firstly, a two-electron reduction of the $C=O$ bond can result in the formation of alcohols. In addition, deoxygenative reduction, also known as hydrogenolysis, can be achieved via a four-electron transfer process, which requires a higher reduction potential. **Thus, controlling hydrogenolysis over hydrogenation is challenging and critical to increasing the yield and selectivity of the desired deoxygenative product.** In present work, we found the application of a titanium catalyst, Zn anode and acidic condition provide an effective route to achieve selective hydrogenolysis. A recent study also supported this result (*Nat. Catal.* **2024**, 7, 43–54).

(2) Regarding the effect of titanium catalysis, we also tested more substrates for the deoxygenative coupling, both in the presence and absence of a titanium catalyst. In addition to the model substrate propionic acid, secondary, tertiary carboxylic acids and benzoic acid led to higher yields in the presence of the titanium catalyst. We think titanium is responsible for the activation of carbonyl group, which is a well-established chemistry in several important reactions such as the McMurry reaction, pinacol coupling, Kulinkovich reaction

and so on. Furthermore, high-resolution mass analysis also provided experimental evidence for the presence of a titanium carbene intermediate as shown in Fig. 4d of the manuscript. These results suggest that titanium indeed facilitates the deoxygenative reductions. **We have included the role of titanium catalyst and revised the inaccurate sentence in the abstract.**

2. In Fig. 3, the authors commented on line 123, ‘Notably, in some cases the aromatic C-H position was also partly deuterated with this method.’, the reason for obtaining these ‘partly deuterated’ results should be provided.

Our response: to demonstrate this result, we have calculated the charge distribution at the aromatic C-H position using **natural population analysis (NPA)**. As shown below, there is a greater charge distribution at the *ortho*-position to the ether functionality compared to the *meta*- and *para*-positions. Consequently, for some substrate in Fig 3a of the manuscript, we observed partial deuteration at the *ortho*-position of ether group. In addition, a similar observation was also reported by T. Werner and co-workers (*Org. Lett.* **2017**, *19*, 5768). **We have mentioned this result in the manuscript on page 6 lines 129-131 and included a discussion of it in the Supporting Information on page S16.**

3. Personally, this reviewer DO NOT like the writing style of abstract in this manuscript, which only comments the potential or specious merits but escapes the key mechanistic insight of the reaction, unless the authors may also believe that this work is not that new and nothing relating to the mechanism is worth to be commented in their abstract.

Our response: according to the comments from this reviewer, we made some revisions:

(1) The abstract is significantly revised to provide more information about the mechanism part.

Updated abstract:

Carboxylic acids are widely available and generally inexpensive from abundant biomass feedstocks, and they are suitable and generic coupling partners in synthetic chemistry. Reported herein is an electroreductive coupling of stable and versatile carboxylic acids with (hetero)arenes using protons as the hydrogen source. **The application of an earth-abundant titanium catalyst has significantly improved the deoxygenative reduction process. Preliminary mechanistic studies provide insights into the deoxygenative reduction of in-situ generated ketone pathway, and the intermediacy generation of ketyl radical and alkylidene titanocene.** Without the necessity of pressurized hydrogen or stoichiometric hydride as reductants, this protocol enables highly selective and straightforward synthesis of various functionalized and structurally diverse alkylbenzenes under mild conditions. The utility of this reaction is further demonstrated through practical and valuable isotope incorporation from readily available deuterium source.

- (2) The mechanism section, as depicted in Fig. 5, has been revised to highlight the ketone *Int-A*, ketyl radical *Int-B*, carbene *Int-C* and alkylidene titanocene intermediates *Int-D*. Additionally, we have included an **alternative pathway in the absence of a titanium catalyst**. The updated mechanism is proposed as follows:

We thank editor and three reviewers' comments and suggestions to improve our manuscript.

REVIEWERS' COMMENTS

Reviewer #1 (Remarks to the Author):

The authors adequately addressed my concerns, so I believe this work is now more appropriate for publication in Nature Communications.

Reviewer #2 (Remarks to the Author):

In the new version, Liu and coworkers have refined the details of experiments and polished the article. The author has satisfactorily addressed all comments. The previously missing parts are well presented in the manuscripts. The quality of the manuscript has improved significantly compared to the previous submission. Thus, we suggest that this article be published.

Reviewer #3 (Remarks to the Author):

The comments made by three reviewers have been carefully responded, and the revised manuscript looks good.